# Knowledge, attitudes, and practices related to COVID-19 among patients attending public dental clinics in Tanzania: A cross-sectional study

Karpal Singh Sohal [1,2¤]*, Rewald L. Moris[3☯], Jeremiah Robert Moshy[2☯]

1 Department of Oral Health Services, Muhimbili National Hospital, Dar es Salaam, Tanzania, 2 Department of Oral and Maxillofacial Surgery, School of Dentistry, Muhimbili University of Health and Allied Sciences, Dar es Salaam, Tanzania, 3 Department of Orthodontics, Paedodontics and Community Dentistry, School of Dentistry, Muhimbili University of Health and Allied Sciences, Dar es Salaam, Tanzania

☯ These authors contributed equally to this work.
¤ Current address: Department of Oral and Maxillofacial Surgery, Muhimbili University of Health and Allied Sciences, Dar es Salaam, Tanzania
* karpal@live.com

**Data Availability Statement:** All relevant data are within the manuscript and its Supporting Information files.

## Abstract

### Introduction

The risk of contracting COVID-19 through dental treatment is potentially high, thus several guidelines have been developed to minimize the spread of COVID-19 in the dental office worldwide. These changes have posed some relevant questions among oral health professionals regarding the understanding and attitudes of dental patients toward COVID-19. This study aimed to assess the knowledge, attitude, and practices of dental patients in Dar es Salaam, Tanzania.

### Methodology

This cross-sectional study was carried out in 4 public hospitals in Dar es Salaam, Tanzania involving 472 adult patients. Data were collected using a questionnaire that had a set of questions regarding knowledge, attitude, and practices (KAP) related to COVID-19. Data were analyzed using the SPSS computer software version 26. For descriptive analysis means, standard error of the mean, and proportion were used. Multivariate regression analysis was utilized for the response analysis. Alpha of less than 0.05 was considered to indicate statistical significance.

### Results

Good levels of knowledge, attitude, and practices related to COVID-19 were found in 76.5%, 74.8%, and 58.1% of participants respectively. On performing multivariate analysis, odds of having good knowledge regarding COVID-19 were almost 2 folds higher in participants who were females, with high education levels, those without partners, and those with stable income. Females were 1.5 folds more likely to have a good attitude toward COVID-19 and odds of good practice against COVID-19 were 3 folds higher in young adults compared to the elderly.

**Funding:** The authors received no specific funding for this work.

**Competing interests:** The authors have declared that no competing interests exist.

## Conclusion

A majority of dental patients have good knowledge and attitude related to COVID-19. Predictors of good knowledge were sex, education level, marital status, and income of the participants. Sex predicted good attitude and age predicted good practice.

## Introduction

The coronavirus disease 2019 (COVID-19) was initially declared a global health emergency on January 30, 2020 [1], and a global pandemic by the World Health Organization (WHO) on March 11, 2020, however, the first case of COVID-19 in Tanzania was confirmed on March 16, 2020 [2] and since then thousands of cases have been diagnosed in the country.

COVID-19 gets transmitted directly from person to person via droplets (that are produced when an infected person coughs, sneezes, or exhales) that reach the nose, mouth, or eyes of another person [1]. It may as well get transmitted indirectly when healthy individuals touch their faces after touching a contaminated surface (by droplets or body fluids from an infected person) [3]. Generally, the risk of cross-infection between the patient and the oral health care provider is high because dental treatment requires the proximity of dental practitioners to patients [4]. Considering the higher chances of airborne transmission of SARS CoV-2 through aerosols, the risk of contracting COVID-19 through dental treatment is potentially high [3].

Several guidelines have been developed to minimize the spread of COVID-19 in dental offices worldwide. They include separating the administration and patient waiting area by a panel (glass or plastic), keeping the waiting room empty without toys and/or magazines, avoiding the overlap of appointments, easy availability of hand sanitizers for the patients, and maintaining distance (6 feet) between one patient and the other [5, 6]. Likewise, oral health professionals are also required to observe protocols of infection control related to dressing, personal protective gear, and disinfection of the office [5].

To determine the patient's readiness to accept changes introduced into the dental practice assessment of their knowledge, attitudes, and practices (KAP) is a fundamental step [7]. Globally, since the outbreak of COVID-19, several studies have been carried out to assess the KAP of oral health professionals related to COVID-19 [1, 4, 8], but there is a paucity of information regarding the same subject among dental patients. Since COVID-19, in a dental setup, may be spread from an infected patient to either another patient or to the dental staff and vice versa, the task of reducing COVID-19 spread is a matter of concern for both the patients and the health professionals. Patients have to observe the guidelines or protocols put in place in several dental clinics. Therefore, it is of paramount importance to assess the dental patients' KAP regarding COVID-19 to get baseline information and therefore plan the best approaches to implement preventive programs and health awareness plans regarding COVID-19. The present study was undertaken to assess the knowledge, attitude, and practices (KAP) related to COVID-19 among patients attending public dental clinics in Tanzania.

## Materials and methods

### Study design, setting, and population

This was a cross-sectional study that involved adult patients who attended treatment in four public dental clinics in Dar-es-Salaam between November 2021 and May 2022. The selected dental clinics were those in Muhimbili National Hospital, the Muhimbili University of Health

and Allied Sciences (MUHAS), and two regional referral hospitals (Mwananyamala, and Temeke).

The inclusion criteria included all dental patients aged 18 and above who visited the selected dental clinics. The exclusion criteria included individuals with intellectual disabilities and patients who were in severe pain, hence could not answer the questions.

The sample size was estimated using the population adjustment formula for single proportion estimation [9]. A sample of 472 participants was calculated based on a 95% confidence level, a 4.5% precision, a power of 0.8, and an expected proportion of 50%. A stratified random sampling method was used, whereby the included hospitals were used as strata. The list of patients who attended the dental clinics on the day of data collection was obtained and each was assigned a number. Using a simple random sampling method, the study participants were thus obtained. Considering the variation in the number of dental patients visiting different dental clinics in the city, dental patients attending MUHAS Dental Clinic contributed 35% of the estimated sample size. Patients from MNH Dental Clinic comprised 25% of the sample, and those from Mwananyamala and Temeke Hospitals Dental Clinics constituted 20% each.

## Questionnaire and data collection

There was no relevant validated study tool available for use in this research setting. Hence, we developed a questionnaire in English (SI-1) which was later translated to Swahili (SI-2). Before using the Swahili version of the questionnaire, it was back-translated into English by an independent translator to check for consistency with the original version. The questionnaire was composed of questions inquiring about the sociodemographic characteristics of the participants (age, sex, marital status, level of education, and occupation), 24 questions on knowledge related to COVID-19 (mode of transmission, symptoms, and preventive measures), 6 questions on the attitude towards COVID-19, and 11 practice related questions. The knowledge and attitude questions were measured by the *"agree/disagree"* format, while practice questions were measured using the *"yes/no"* format. Each correct response was allocated 1 point. Reliability was assessed by carrying out a pilot study. The questionnaire was distributed to 25 subjects and again the same group of individuals was asked to respond to the questionnaire after 10 days. Cohen's Kappa statistics was performed with an almost perfect agreement in responses [K = 0.84 (95% CI, 0.54–1.14), p<0.001].

During data collection, the Swahili version of the questionnaire was distributed to the patients while they awaited in the waiting area of the dental clinics. The investigator was always around to keep an eye on patients to ensure they did not discuss the answers to the questions. For a few patients without formal education, the investigator interviewed them and recorded their responses to the questionnaire. There was no follow-up visit of the patients who had taken part in this study.

## Statistical analysis

The Statistical Package for Social Sciences software (SPSS) for Windows (version 26, Armonk, New York: IBM Corp) was used to coded and analyzed the data. For descriptive analysis means, standard error of the mean, and proportion were used.

The participants' age was categorized as young adults (< 40 years), middle-aged (40–59 years), and elderly (≥ 60 years). The level of education was dichotomized into a low level (no formal and primary education) and a high level (secondary and tertiary education). Marital status was grouped into those with partners (married, cohabiting) and those without partners (single, divorced, widowed). Employment status was grouped into stable income (civil servants

and private formal employment) and unstable income (informal employment, peasants, students, unemployed, and retired).

The cut-off point between good and poor knowledge, attitude, and practice domains was decided based on the sample mean points for each domain. The cut-off points for good were $\geq 18$ points, $\geq 5$ points, and $\geq 8$ points for knowledge, attitude, and practices respectively.

Univariate analysis was carried out to assess factors associated with good knowledge, attitude, and practices related to COVID-19. The probability level of $\alpha < 0.05$ was selected for statistical significance. Multivariate logistic regression was used to assess the strength of the association between good knowledge, attitude, and practices related to COVID-19 and the predictor variables.

### Ethics statement

Ethical clearance was sought from the MUHAS research and ethics committee (DA.25/111/01B/99), The authorities of MUHAS Dental Clinic, MNH, Temeke, and Mwananyamala Hospitals provided permission to conduct the study in their settings. Only those participants who freely gave consent to participate were included in the study. Consent was ensured when a consent form was signed at an area designated that the consent was granted. All information was handled confidentially and refusal to participate or withdraw from the study did not result in any consequence.

## Results

### Characteristics of the study population

Among 472 participants, females were 241 (51.1%) with male to female ratio of 1:1.04. Their age ranged between 18 years and 85 years. The mean age was 33.7 years (SEM = 0.58). A majority (N = 344; 72.9%) were young adults. Most (N = 195; 41.3%) participants had attained a secondary level of education. Nearly half (N = 227; 48.1%) of the participants were single and most (N = 150; 31.8%) were vendors (Table 1).

### Knowledge regarding COVID-19

The mean knowledge score was 18.9 points (SEM = 0.12) out of a total possible score of 24 points (range 2–24 points). A majority (N = 447; 94.7%) of participants knew that COVID-19 is transmitted through air droplets. A few (N = 79; 37.9%) participants agreed that a person infected with COVID-19 can remain asymptomatic. The majority (> 90%) of participants agreed that the common symptoms of COVID-19 are fever, fatigue, and difficulty in breathing. Nealy all participants agreed that washing hands (N = 495; 98.5%), using hand sanitizers (N = 453; 96%), and wearing masks (N = 444; 94.1%) are preventive measures against COVID-19 infection (Table 2).

### Attitude towards COVID-19

The mean attitude score was 5.1 points (SEM = 0.06) out of 6 points (range 0–6 points). A majority (N = 448; 94.9%) of participants agreed that COVID-19 was a life-threatening disease and 460 (97.5%) agreed that health education has a role to play in the control of COVID-19 spread. However, only about two-thirds (N = 314, 66.5%) agreed that it is important to be vaccinated against COVID-19 (Table 2).

**Table 1. Distribution of study participants according to sociodemographic characteristics.**

| Sociodemographic characteristic | | Frequency (N) | Percentage (%) |
|---|---|---|---|
| Age group (year) | 18–39 | 343 | 72.7 |
| | 40–59 | 104 | 22.0 |
| | 60+ | 25 | 5.3 |
| Sex | Male | 231 | 48.9 |
| | Female | 241 | 51.1 |
| Education level | Informal education | 4 | 0.8 |
| | Primary | 108 | 22.9 |
| | Secondary | 195 | 41.3 |
| | Tertiary | 165 | 35.0 |
| Marital status | Single | 227 | 48.1 |
| | Married | 197 | 41.7 |
| | Cohabiting | 29 | 6.1 |
| | Divorced | 6 | 1.3 |
| | Widowed | 13 | 2.8 |
| Occupation | Civil servant | 45 | 9.5 |
| | Private formal sector | 96 | 20.3 |
| | Informal employment | 150 | 31.8 |
| | Peasant | 51 | 10.8 |
| | Student | 66 | 14.0 |
| | Unemployed/retired | 64 | 13.6 |

## Practices against COVID-19

The mean practice score was 7.6 points (SEM = 0.1) of a total possible score of 11 points (range 2–11 points). The common practices against COVID-19 were hand hygiene (N = 431; 91.3%) and wearing face masks in public places (N = 124; 26.3%). The least preventive measure was the use of herbal remedies (Table 3).

## Factors associated with knowledge, attitudes, and practices regarding COVID-19

Good levels of knowledge, attitude, and practices related to COVID-19 were found in 76.5%, 74.8%, and 58.1% of participants respectively. Upon performing univariate analysis sex, education level, marital status, and income were significantly associated with knowledge (p < 0.05). Whereas only the sex of the participant was significantly associated with attitude (p = 0.038) and both sex and age group of the participant were associated with practices (p<0.05) (Table 4).

Multiple regression analysis showed that odds of having good knowledge regarding COVID-19 were almost 2 folds higher in participants who were females, with high education levels, those without partners, and those with stable income. The chances of females having a good attitude toward COVID-19 were 1.5 folds higher than males, and young adults were nearly 3 times more likely to have good practice against COVID-19 compared to the elderly (Table 5).

## Discussion

In routine dental practice risks of cross-infection are significant considering infectious diseases (including COVID-19) can be transmitted either directly (contact with blood, oral fluids, and

**Table 2. Distribution of study participants according to preferred response to the question regarding knowledge and attitude related to COVID-19.**

| Questions regarding different aspects of COVID-19 | Preferred response | Frequency (N) | Percentage (%) |
|---|---|---|---|
| **Knowledge regarding COVID-19** | | | |
| **Are the following common modes of COVID-19 transmission?** | | | |
| 1. Air droplets | Agree | 447 | 94.7 |
| 2. Indirect contact | Disagree | 136 | 28.8 |
| 3. Body fluid | Disagree | 177 | 37.5 |
| 4. Aerosols | Agree | 406 | 86.0 |
| 5. Sexual | Disagree | 380 | 80.5 |
| **Are the following common symptoms of COVID-19?** | | | |
| 6. Fever | Agree | 431 | 91.3 |
| 7. Fatigue | Agree | 431 | 91.3 |
| 8. Limb edema | Disagree | 429 | 90.9 |
| 9. Headache | Agree | 396 | 83.9 |
| 10. Alopecia | Disagree | 448 | 94.9 |
| 11. Loss of smell and taste sensation | Agree | 157 | 33.3 |
| 12. Vomiting | Disagree | 450 | 95.3 |
| 13. Flu | Agree | 399 | 84.5 |
| 14. Difficulty in breathing | Agree | 440 | 93.2 |
| 15. Nasal bleeding | Disagree | 383 | 81.1 |
| 16. Dry cough | Agree | 334 | 70.8 |
| 17. Diarrhea | Disagree | 394 | 83.5 |
| 18. It can remain asymptomatic | Agree | 179 | 37.9 |
| **Are the following preventive measures against COVID-19?** | | | |
| 19. Washing hands with soap and running water. | Agree | 465 | 98.5 |
| 20. Use of sanitizer | Agree | 453 | 96.0 |
| 21. Frequent touching of nose, eyes, and mouth. | Disagree | 371 | 78.6 |
| 22. Wearing masks in public places. | Agree | 444 | 94.1 |
| 23. Maintaining social distance | Agree | 434 | 91.9 |
| 24. Taking bath twice a day | Disagree | 335 | 71.0 |
| **Attitude towards COVID-19** | | | |
| 1. Do you think COVID-19 is a life-threatening condition? | Yes | 448 | 94.9 |
| 2. Do you think complying with the precaution measures introduced by the World Health Organization will prevent the spread of COVID-19? | Yes | 397 | 84.1 |
| 3. Do you think it is important for people to be vaccinated against COVID-19? | Yes | 314 | 66.5 |
| 4. Do you think that it is necessary to have a general screening for COVID-19 (e.g.measuring body temperature) during a regular dental checkup? | Yes | 348 | 73.7 |
| 5. Do you agree that self-protection against COVID-19 is necessary to protect others? | Yes | 424 | 89.8 |
| 6. Do you think that health education can play an important role to control COVID-19? | Yes | 460 | 97.5 |

other body secretions) or by indirect contact (contaminated instruments and environmental surfaces) [4, 10, 11]. Because of this, efforts to prevent cross-infection between oral healthcare professionals and patients as well as between patients themselves are inevitably crucial. This can be achieved by adhering to the universally recommended guidelines for preventing cross-infection in dental practice [10].

Infection control practices in dentistry during the era of COVID-19 have been of high priority, and as per WHO recommendations some additional measures like maintaining hand hygiene, use of alcohol-based hand sanitizers, face masks, social distancing, and getting body temperature checked are implemented [7]. The oral health professionals are playing their part,

**Table 3. Distribution of study participants according to their practice related to COVID-19.**

| Practice against COVID-19 (within the past one week) | Preferred response | Frequency (N) | Percentage (%) |
|---|---|---|---|
| 1. I have been washing my hands with tap water and soap. | Yes | 445 | 94.3 |
| 2. I have been using hand sanitizer. | Yes | 400 | 84.7 |
| 3. I wear masks in public places. | Yes | 431 | 91.3 |
| 4. I have been using antibiotics. | No | 350 | 74.2 |
| 5. I shower immediately on getting home from work. | Yes | 264 | 55.9 |
| 6. I have been eating citrus fruits, ginger, and vitamin c for boosting immunity | Yes | 364 | 77.1 |
| 7. I use herbal remedies. | Yes | 124 | 26.3 |
| 8. I use a handkerchief while coughing and/or sneezing. | Yes | 324 | 68.6 |
| 9. I do regular physical activity. | Yes | 297 | 62.9 |
| 10. I have been shaking hands with others. | No | 307 | 65.0 |
| 11. I have been touching my eyes, nose, and mouth before washing my hands. | No | 303 | 64.2 |

but for the patients has been a new practice. This calls for assessing patients' knowledge, attitude, and practices related to COVID-19 to understand their readiness to accept newly introduced protocols.

The results of the current study revealed that 76.5% of the study participants had a good level of knowledge regarding COVID-19 with respect to the common mode of transmission, symptoms, and preventive methods. This was slightly lower than the findings from an online study carried out recently in Tanzania that reported 84.4% of the participants had good knowledge [12]. This difference can be attributed to methodological differences and study population. The previous study was online based thus only educated people and those who had internet access did participate, unlike the current study in which dental patients from all walks of life took part. Despite the slight difference noted between the results of these studies, it is worth generalizing that the population in Tanzania has sufficient knowledge regarding COVID-19. This indicates the efforts put by the Ministry of Health to sensitize citizens via

**Table 4. Univariate analysis of factors associated with knowledge, attitudes, and practices related to COVID-19.**

| Socio-demographic characteristics | | Knowledge | | Attitude | | Practice | |
|---|---|---|---|---|---|---|---|
| | | Poor | Good | Poor | Good | Poor | Good |
| Age group (year) | <40 | 75 (21.9%) | 268 (78.1%) | 80 (23.3%) | 263 (76.7%) | 135 (39.4%) | 208 (60.6%) |
| | 40–59 | 28 (26.9%) | 76 (73.1%) | 30 (28.8%) | 74 (71.2%) | 47 (45.2%) | 57 (54.8%) |
| | 60+ | 8 (32.0%) | 17 (68.0%) | 9 (36.0%) | 16 (64.0%) | 16 (64.0%) | 9 (36.0%) |
| *p-value* | | *0.334* | | *0.232* | | *0.041* | |
| Sex | Female | 47 (19.5%) | 194 (80.5%) | 51 (21.2%) | 190 (78.8%) | 91 (37.8%) | 150 (62.2%) |
| | Male | 64 (27.7%) | 167 (72.3%) | 68 (29.4%) | 163 (70.6%) | 107 (46.3%) | 124 (53.7%) |
| *p-value* | | *0.036* | | *0.038* | | *0.06* | |
| Education level | Low level | 42 (37.5%) | 70 (62.5%) | 36 (32.1%) | 76 (67.9%) | 54 (48.2%) | 58 (51.8%) |
| | High level | 69 (19.2%) | 291 (80.8%) | 83 (23.1%) | 277 (76.9%) | 144 (40.0%) | 216 (60.0%) |
| *p-value* | | *<0.001* | | *0.053* | | *0.124* | |
| Marital status | With a partner | 65 (28.8%) | 161 (71.2%) | 64 (28.3%) | 162 (71.7%) | 93 (41.2%) | 133 (58.8%) |
| | Without a partner | 46 (18.7%) | 200 (81.3%) | 55 (22.4%) | 191 (77.6%) | 105 (42.7%) | 141 (57.3%) |
| *p-value* | | *0.01* | | *0.136* | | *0.736* | |
| Occupation | Stable income | 23 (16.3%) | 118 (83.7%) | 36 (25.5%) | 105 (74.5%) | 63 (44.7%) | 78 (55.3%) |
| | Unstable income | 88 (26.6%) | 243 (73.4%) | 83 (25.1%) | 248 (74.9%) | 135 (40.8%) | 196 (59.2%) |
| *p-value* | | *0.016* | | *0.917* | | *0.432* | |

**Table 5. Multivariate analysis of factors associated with knowledge, attitudes, and practices related to COVID-19.**

| Socio-demographic characteristics | | Adjusted Odds Ratio | | |
| --- | --- | --- | --- | --- |
| | | Knowledge AOR (95%CI) | Attitude AOR (95%CI) | Practice AOR (95%CI) |
| Age group (year) | <40 | 1.42 (0.58–3.51) | 1.67 (0.71–3.99) | 2.79 (1.18–6.58) |
| | 40–59 | 1.23 (0.46–3.29) | 1.4 (0.55–3.59) | 2.19 (0.87–5.52) |
| | 60+ | 1 (Ref.) | 1 (Ref.) | 1 (Ref.) |
| Sex | Female | 1.75 (1.12–2.74) | 1.56 (1.02–2.39) | 1.4 (0.96–2.03) |
| | Male | 1 (Ref.) | 1 (Ref.) | 1 (Ref.) |
| Education level | Low level | 1 (Ref.) | 1 (Ref.) | 1 (Ref.) |
| | High level | 2.05 (1.25–3.36) | 1.5 (0.92–2.45) | 1.46 (0.93–2.3) |
| Marital status | With a partner | 1 (Ref.) | 1 (Ref.) | 1 (Ref.) |
| | Without a partner | 1.82 (1.13–2.93) | 1.25 (0.80–1.97) | 0.79 (0.53–1.18) |
| Occupation | Stable income | 2.09 (1.19–3.67) | 1.04 (0.63–1.70) | 0.8 (0.52–1.24) |
| | Unstable income | 1 (Ref.) | 1 (Ref.) | 1 (Ref.) |

various media outlets regarding the pandemic were fruitful. Regardless of good knowledge among dental patients, it was slightly worrying that most of them did not know that a person with COVID-19 may be asymptomatic.

Similar to the results of a study by Lee et al. [13] the findings from the current study show that males and less educated participants had low knowledge related to COVID-19 than their counterparts. In this study the odds of having good knowledge regarding COVID-19 were almost 2 folds higher in participants who were females, with high education levels, those without partners, and those with stable income. In one study, it has been pointed out that females are more equipped with knowledge related to COVID-19 because they use social media more than their male counterparts to gather knowledge on COVID-19 [14], the same explanation may be true in our setting as well. In line with findings in the literature [12, 15, 16] participants who were single and those with higher levels of education had greater odds of having good knowledge related to COVID-19 than their counterparts, and it was suggested that singles and educated have better access to information [15].

Good attitude related to COVID-19 was noted in almost three-quarters of the participants of this study and this was relatively higher than the attitude of the population elsewhere [7, 17]. The attitude did differ significantly by sex with the chances of females having a good attitude toward COVID-19 being 1.5 folds higher than male participants. This was contrary to findings from Bangladesh where the attitude related to COVID-19 differed by age, education, marital status, and monthly income of the participants [17]. Poor attitude in males compared to females may be due to their general unwillingness and low motivation to engage with health-related information [18].

An encouraging finding of the current study is that 66% of the participants thought the vaccine against COVID-19 was important. This finding was comparably higher than what was reported in India [19] regarding attitude toward the importance of the vaccine. The slightly above-average acceptability of the vaccine against COVID-19 may be attributed to not having enough information regarding the vaccine as far as its safety standards and long-term effects are of concern [20].

Despite a majority of participants having a good level of knowledge and positive attitude related to COVID-19, the practices against the pandemic were fairly low similar to findings from Malaysia [21]. The low level of practice was also reported by Singh et al. [7], however, the findings of Bains et al. [3] indicated good practice in about 80% of the participants. The finding from this study does indicate that having good knowledge and attitude does not necessarily

warrant good practice. Though Singh et al. [7] found a significant positive correlation between attitude and practices to the contrary, in a study from China low correlation was found between knowledge and practices, and no correlation was found between attitude and practices [21].

In the current study young adults were nearly 3 times more likely to have good practice against COVID-19 compared to the elderly. Considering the elderly are at high risk to contract the disease due to chronic diseases, malnutrition, drug use, impaired cognition, and social factors which contribute to declining immune function [22], it was expected they would have better practices against COVID-19. The better practice among young adults may have been contributed by their ability to access information easily.

## Limitation

The study has some limitations that require to be pointed out. First, it was a cross-sectional study hence, may not establish the causal inferences. Second, the participants were asked to report their practices against COVID-19 within a week, their responses might have been subject to recall bias. Finally, we used close-ended questions for assessing the knowledge, attitude, and practices, which by virtue of its design gave limited information. In addition, the question regarding the source of information was not taken into account in this study. Despite these limitations, the results from this study carry valuable information about the KAP of patients in Tanzania.

## Conclusion

The findings of the study, therefore, indicate that majority of the dental patients who attend public dental clinics in Dar es Salaam have a good level of knowledge and attitude related to COVID-19, and most had good practices against the disease. Good knowledge was associated with the sex, education level, marital status, and income of the participants, whereas good attitude was associated with sex and good practice was related to the age of the participant.

## Supporting information

**S1 File.**
(DOCX)

**S2 File.**
(DOCX)

## Author Contributions

**Conceptualization:** Karpal Singh Sohal, Rewald L. Moris.

**Data curation:** Karpal Singh Sohal, Rewald L. Moris, Jeremiah Robert Moshy.

**Formal analysis:** Karpal Singh Sohal.

**Investigation:** Rewald L. Moris.

**Methodology:** Karpal Singh Sohal, Rewald L. Moris, Jeremiah Robert Moshy.

**Project administration:** Karpal Singh Sohal.

**Resources:** Karpal Singh Sohal, Jeremiah Robert Moshy.

**Supervision:** Karpal Singh Sohal, Jeremiah Robert Moshy.

**Validation:** Jeremiah Robert Moshy.

**Writing – original draft:** Karpal Singh Sohal, Rewald L. Moris.

**Writing – review & editing:** Karpal Singh Sohal, Jeremiah Robert Moshy.

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
