## [Decision Letter · Decision Letter 0]

9 Sep 2022

PONE-D-22-21911Knowledge, attitudes, and practices related to COVID-19 among patients attending public dental clinics in Tanzania: A cross-sectional studyPLOS ONE

Dear Dr. Karpal Singh Sohal,

Thank you for submitting your manuscript to PLOS ONE. After careful consideration, we feel that it has merit but does not fully meet PLOS ONE’s publication criteria as it currently stands. Therefore, we invite you to submit a revised version of the manuscript that addresses the points raised during the review process.

We look forward to receiving your revised manuscript.

Kind regards,

Kehinde Kazeem Kanmodi, BDS

Academic Editor

PLOS ONE

Journal Requirements:

a) Did participants provide their written or verbal informed consent to participate in this study?

3. Thank you for providing the English version of the questionnaire used in your study. Please also include the Swahili version of the questionnaire as supplementary file.

Additional Editor Comments:

Most of your references do not have journal issue number; where appropriate, they must be included. Also, ensure that you strictly follow the manuscript preparation guidelines of PLOS ONE--this is one of the conditions you must meet before your manuscript can be considered acceptable for publication.

Reviewers' comments:

Reviewer's Responses to Questions

**Comments to the Author**

1. Is the manuscript technically sound, and do the data support the conclusions?

Reviewer #1: Yes

Reviewer #2: Partly

2. Has the statistical analysis been performed appropriately and rigorously? 

Reviewer #1: Yes

Reviewer #2: Yes

3. Have the authors made all data underlying the findings in their manuscript fully available?

Reviewer #1: Yes

Reviewer #2: Yes

4. Is the manuscript presented in an intelligible fashion and written in standard English?

Reviewer #1: Yes

Reviewer #2: Yes

5. Review Comments to the Author

Reviewer #1: The article is well laid out and gives a great insight into the subject matter in Tanzania.

The following should however be resolved.

1. Table 2 has disagree as the preferred option for body fluid and indirect contact as mode of transmissions, I think this should be reviewed.

2. Line 230 does not convey a clear idea

3. line 231 the discussion here has to be worked on. the author submission insinuates that the vaccine should be discouraged. I believe this is not the intent.

4. Line 244, the author should elucidate more on the risks in the elderlies and how to mitigate these risks.

5. line 255, dental is not necessary and should be deleted.

Reviewer #2: I have read this paper carefully and reached the following conclusions;

The paper is relevant and could be considered for publication.

Authors should however, expatiate on the mode of administration of the questionnaires and the assessment of reliability as stated in the methodology. Authors also did not state if there was a follow-up visit and this needs to be clarified. Furthermore, authors should state clearly if there was a pilot survey.

6. PLOS authors have the option to publish the peer review history of their article (what does this mean?). If published, this will include your full peer review and any attached files.

Reviewer #1: No

Reviewer #2: No

---

## [Author Response · Author response to Decision Letter 0]

12 Sep 2022

We are very grateful to the entire Editorial Team of the journal and to the reviewers for the hard work you have done to ensure our manuscript entitled “Knowledge, attitudes, and practices related to COVID-19 among patients attending public dental clinics in Tanzania: A cross-sectional study

” is of high quality.

We have read and taken into consideration the constructive comments of both the academic editor and the reviewers. We have made all necessary amendments to our work. Track changes have been instituted in the manuscript in response to the editor and the reviewers’ comments.

---

## [Decision Letter · Decision Letter 1]

11 Oct 2022

Knowledge, attitudes, and practices related to COVID-19 among patients attending public dental clinics in Tanzania: A cross-sectional study

PONE-D-22-21911R1

Dear Dr. Karpal Singh Sohal,

We’re pleased to inform you that your manuscript has been judged scientifically suitable for publication and will be formally accepted for publication once it meets all outstanding technical requirements.

Kind regards,

Kehinde Kazeem Kanmodi, BDS

Academic Editor

PLOS ONE

Additional Editor Comments (optional):

Nil.

Reviewers' comments:

Reviewer's Responses to Questions

**Comments to the Author**

1. If the authors have adequately addressed your comments raised in a previous round of review and you feel that this manuscript is now acceptable for publication, you may indicate that here to bypass the “Comments to the Author” section, enter your conflict of interest statement in the “Confidential to Editor” section, and submit your "Accept" recommendation.

Reviewer #2: All comments have been addressed

2. Is the manuscript technically sound, and do the data support the conclusions?

Reviewer #2: Yes

3. Has the statistical analysis been performed appropriately and rigorously? 

Reviewer #2: Yes

4. Have the authors made all data underlying the findings in their manuscript fully available?

Reviewer #2: Yes

5. Is the manuscript presented in an intelligible fashion and written in standard English?

Reviewer #2: Yes

6. Review Comments to the Author

Reviewer #2: Authors have adequately addressed my comments. The manuscript was written clearly in standard English. I think this paper may be considered for publication.

7. PLOS authors have the option to publish the peer review history of their article (what does this mean?). If published, this will include your full peer review and any attached files.

Reviewer #2: No

---

## [Editor Report · Acceptance letter]

18 Oct 2022

PONE-D-22-21911R1 

Knowledge, attitudes, and practices related to COVID-19 among patients attending public dental clinics in Tanzania: A cross-sectional study 

Dear Dr. Sohal:

I'm pleased to inform you that your manuscript has been deemed suitable for publication in PLOS ONE. Congratulations! Your manuscript is now with our production department. 

Kind regards, 

on behalf of

Dr. Kehinde Kazeem Kanmodi 

Academic Editor

PLOS ONE